# Review and Updates on the Diagnosis of Tuberculosis

**DOI:** 10.3390/jcm11195826

**Published:** 2022-09-30

**Authors:** Yi Huang, Lin Ai, Xiaochen Wang, Ziyong Sun, Feng Wang

**Affiliations:** Department of Laboratory Medicine, Tongji Hospital, Tongji Medical College, Huazhong University of Science and Technology, Wuhan 430030, China

**Keywords:** tuberculosis, diagnosis, laboratory methods, innovative techniques

## Abstract

Diagnosis of tuberculosis, and especially the diagnosis of extrapulmonary tuberculosis, still faces challenges in clinical practice. There are several reasons for this. Methods based on the detection of *Mycobacterium tuberculosis* (Mtb) are insufficiently sensitive, methods based on the detection of Mtb-specific immune responses cannot always differentiate active disease from latent infection, and some of the serological markers of infection with Mtb are insufficiently specific to differentiate tuberculosis from other inflammatory diseases. New tools based on technologies such as flow cytometry, mass spectrometry, high-throughput sequencing, and artificial intelligence have the potential to solve this dilemma. The aim of this review was to provide an updated overview of current efforts to optimize classical diagnostic methods, as well as new molecular and other methodologies, for accurate diagnosis of patients with Mtb infection.

## 1. Introduction

Tuberculosis (TB) is a leading global public health problem, with high morbidity and mortality in humans. Until the COVID-19 pandemic, TB was still the leading cause of death from a single infectious agent, ranking above HIV/acquired immune deficiency syndrome [1]. The number of people newly diagnosed with TB fell from 7.1 million in 2019 to 5.8 million in 2020, and reduced access to TB diagnosis and treatment has resulted in an increase in TB deaths [1]. Generally, although progress has been made in reducing the TB burden worldwide, this has been insufficient to reach the first milestone of the End TB Strategy.

Drug-resistant tuberculosis is another threat to the global control of the disease. The emergence of multi-drug resistant tuberculosis (MDR-TB) and extensively drug resistant tuberculosis (XDR-TB) is increasing in some regions of the world. Globally in 2020, 71% (2.1/3.0 million) of people diagnosed with bacteriologically confirmed pulmonary TB were tested for rifampicin resistance, with considerable variation among countries [1]. Among them, 132,222 cases of MDR-TB and 25,681 cases of pre-XDR-TB or XDR-TB were detected [1]. The increase of drug-resistant TB demands accurate and reproducible drug susceptibility testing (DST) methods.

In 2015, the World Health Organization (WHO) proposed a paradigm shift, from a stop TB to end TB strategy [2]. To achieve this, the WHO has emphasized the essential role of early, rapid, and accurate identification of *Mycobacterium tuberculosis* (Mtb) and the determination of drug susceptibility in the treatment and management of this disease. In this article, we provide an overview of current diagnostic methods, which not only focus on Mtb identification but also on the detection of Mtb-specific host response and new techniques for the diagnosis of TB.

## 2. Mtb Diagnosis

### 2.1. Microscopy

Sputum smear microscopy still remains one of the basic methods for identifying Mtb in developing countries. The most regular practice is acid-fast staining using carbol fuschin solution. The lipid-rich cell wall of Mtb resists decolorization with acid-containing reagents, which means that acid-fast organisms can be visualized on microscopic examination of smears prepared from sputum, alveolar lavage fluid, or other specimens. The major limitation of smear microscopy is a lack of sensitivity, which varies widely (20 to 80%) in different studies and is particularly poor in paucibacillary TB, including child TB, extrapulmonary TB, or HIV coinfected TB [3,4,5,6,7,8]. Specificity is likely to vary considerably depending on the local prevalence of infections with nontuberculous mycobacteria (NTM). In regions with a low incidence of NTM, the specificity of smear microscopy can reach up to 98% [9].

Fluorescence microscopy can save manpower and improve work efficiency [3], as well as increasing the sensitivity of smear microscopy [10]. The possibility of false-positive results is a potential shortcoming of fluorescence microscopy, because of non-specific fluorochrome dye incorporation [11]. The instability of fluorescent staining has been reported [12]. Unlike conventional microscopy using conventional artificial light, fluorescence microscopy uses an intense light source, such as a halogen or high-pressure mercury vapor lamp, which is expensive and vulnerable [13]. However, in comparison to intense light sources, light-emitting diodes (LEDs) are more robust, sustainable, and have a longer lasting battery life expectancy, and these qualities make LED microscopy feasible for use in resource-limited settings. Accordingly, the WHO recommends that conventional fluorescence microscopy can be replaced by LED microscopy [10].

Fluorescein diacetate (FDA) is a new stain solution where only living cells actively convert the nonfluorescent FDA into the green fluorescent compound following enzymatic activity [14]. FDA staining can be used to detect the viable Mtb and simply predict the quantitative culture results within 1 h, indicating whether patients are responding to TB therapy [15]. “TBDx” is an innovative smear microscopy system that automatically loads slides onto a microscope, focuses and digitally captures images, and then classifies smears as positive or negative using computerized algorithms [16]. Despite showing potential for detection of Mtb, these new microscopy methods still need more validation of their performance in clinical practice.

Sputum smear microscopy is relatively fast, inexpensive, and specific for Mtb diagnosis in high-TB burden areas. Thus, it is still a worthwhile method for Mtb diagnosis, especially in resource-limited countries. The most important limitation of microscopy is a low sensitivity for diagnosis of TB, especially in paucibacillary samples. In addition, routine microscopy cannot differentiate between live and dead bacilli, which leads to the difficulty of the method for early detection of treatment failure or drug resistance. Generally, the performance of smear microscopy is far from the current needs for diagnosis of TB in clinical practice.

### 2.2. Culture

#### 2.2.1. Solid and Liquid Culture

Culture is still the WHO-recommended gold standard for the diagnosis of TB, as Mtb isolation is not only important for disease diagnosis but also permits the detection of drug resistances. Traditional Mtb culture can be performed on either a solid (e.g., Lowenstein–Jensen) or liquid (e.g., Middlebrook 7H9) medium. Notably, solid culture is less expensive than liquid culture and less prone to contamination by other bacteria or fungi, but liquid culture is faster, more sensitive, and convenient (growth is detected automatically) [17,18].

A technical detail that should be mentioned is sample decontamination. Samples (such as sputum) that are contaminated with normal flora must undergo decontamination before culture. The laboratory can use a routine decontamination reagent such as NaOH together with N-acetyl-L-cysteine (NALC), which kills rapidly growing bacteria and fungi but has a limited effect on Mtb growth [19]. The laboratory should determine the optimal concentration of decontamination reagent, to avoid over-decontamination (which reduces the yield of Mtb) and under-decontamination (which leads to failed cultures, because of high rates of bacterial or fungal growth).

#### 2.2.2. Rapid Identification from Positive Cultures

Rapid identification assays capable of distinguishing between Mtb complex and NTM after positive cultures are the basis for initiating early anti-TB therapy. Traditional biochemical assays are slow and have a long turnaround time (2–3 weeks) [20,21]. Mtb protein 64 (MPT-64) is one of the Mtb-specific antigens secreted during bacterial growth. Immunochromatographic (ICT) assays are based on the principle of a double-sandwich enzyme-linked immunosorbent assay, which detects MPT-64 antigen. A recent review reported a high sensitivity (range, 98.1 to 98.6%) and high specificity (range, 99.2 to 100%) of ICT assays for rapid identification of Mtb complex [22]. In addition, ICT assays are rapid, simple, and without the need of additional special equipment. Therefore, the WHO recommends using ICT assays for rapid identification of Mtb complex from positive cultures [23].

#### 2.2.3. Phenotypic Tests for DST

Testing on solid agar using the proportion method is still regarded as the reference standard method for DST of Mtb, which is performed by counting the number of Mtb colonies that grow on agar with or without antibiotics. The absolute concentration method is based on the comparison of growth intensity in the presence of cutoff concentrations and on drugfree controls. Commercial automated liquid culture systems (e.g., the mycobacteria growth indicator tube system) use a modification of the proportion method and offer reliable results for two important first-line drugs (isoniazid (INH) and rifampin (RIF)), while the testing for resistance to second-line drugs is less reliable and reproducible [9,24].

Other methods have also been reported, such as microscopic-observation drug susceptibility assays (using inverted microscope to observe the characteristic spiral or comma shaped microcolonies of growing mycobacteria in liquid culture), thin-layer agar (identification of isolates based on the characteristic morphology of mycobacteria in solid culture), and colorimetric redox indicator (observation by a change in the color of the culture medium containing anti-TB drugs) [25,26,27]. The current data suggest that these assays could be used for a rapid and accurate DST, in settings where WHO-endorsed assays are not available [28].

Generally, the sensitivity of Mtb culture is higher than smear, and Mtb culture is still one of the most important methods for diagnosis of TB. However, false-negative results of Mtb culture are inevitable in real clinical practice, and using the method for diagnosing TB in paucibacillary samples is still challenging, leading to its inability to rule out TB. Another limitation is that the growth of Mtb in conventional mediums takes from 4 to 8 weeks, with an additional 4 weeks for DST using the conventional proportion method [29]. In addition, Mtb culture requires biosafety facilities and specially trained laboratory technicians to perform the experiment. Hence, Mtb culture is recommended to be performed at national reference or central laboratories, in some areas.

### 2.3. Molecular Tests

#### 2.3.1. Xpert MTB/RIF

Xpert MTB/RIF (Cepheid), an automated molecular test for detection of Mtb and RIF resistance directly from clinical specimens, is one of the most commonly used molecular tests for diagnosis of TB worldwide. It is based on a hemi-nested real-time polymerase-chain-reaction (PCR) assay to amplify an Mtb-specific sequence of the rpoB gene. The turnaround time of this assay is short (2–3 h), and the problem of cross-contamination is eliminated because of self-contained cartridges [30]. It uses three specific primers and five unique molecular probes to ensure a high degree of specificity, and NTM does not confound testing [31].

The sensitivity of Xpert MTB/RIF is 99.8% for smear- and culture-positive cases and 90.2% for smear-negative but culture-positive cases, and the estimated specificity is 99.2% for a single direct MTB/RIF test [32]. As compared with phenotypic DST, the MTB/RIF test correctly identifies RIF resistance with 97.6% sensitivity and 98.1% specificity [32]. Notably, there is some proportion of Xpert MTB/RIF positive results in culture-negative cases, which still should be diagnosed as TB because of the high specificity of this assay [33,34,35,36]. It is worth noting that INH is not detected by this test, while INH resistance accounts for part of the first-line treatment failures [37]. Similar to smear microscopy, both live and dead bacilli are detected by Xpert MTB/RIF, making this test incapable of assessing post-therapy efficacy in the current format [34,38].

Xpert Ultra, the next generation of Xpert MTB/RIF, has a larger chamber for DNA amplification than Xpert MTB/RIF. This new assay has two multicopy amplification targets for TB, namely, IS6110 and IS1081, and therefore allows a lower detection limit than Xpert MTB/RIF [39]. These modifications have increased Ultra’s overall sensitivity from 83% to 88%, with a slight decrease in specificity from 98% to 96% [40]. The WHO recommended Xpert Ultra as the initial TB diagnostic test for adults and children, regardless of HIV status, in 2017 [41]. Another PCR-based cartridge, Xpert MTB XDR-TB, has been designed for detection of mutations associated with resistance to multiple first- and second-line TB drugs or XDR-TB. In 2020, the Xpert MTB XDR-TB cartridge was launched, but further clinical validation is needed [42].

#### 2.3.2. Loop-Mediated Isothermal Amplification (LAMP)

LAMP is an isothermal PCR amplification technique, and the reaction process proceeds at a constant temperature using an auto-cycling strand displacement reaction targeted at the six regions of the gyrB and 16S rRNA genes [43]. The sensitivity of TB-LAMP is slightly lower than that of Xpert MTB/RIF, while the specificity of the two methods is comparable [44]. Nevertheless, TB-LAMP does not require sophisticated laboratory equipment and can be performed in peripheral settings, contributing to its use as a simple, rapid, specific, and cost-effective nucleic acid amplification method [43]. Currently, the TB-LAMP assay is recommended by the WHO as a potential replacement for smear microscopy, due to its superior diagnostic performance [45].

#### 2.3.3. Line Probe Assay (LPA)

LPA detects TB DNA and genetic mutations associated with drug resistance, after DNA extraction and PCR amplification. The basis of the LPA is that the pre-labeled amplification product is captured by the DNA probe solidified on the membrane strip and detected by colorimetry, and the results of LPA appear as a linear band [46]. LPA can detect drug resistance to first-line TB drugs (INH and RIF), and there are different versions of commercial products, including GenoType MTBDRplus 1.0 (Hain Lifescience) and INNO-LiPA Rif TB kit (Innogenetics) [46,47]. The newer generation of LPA, GenoType MTBDRplus 2.0, is more sensitive for the detection of Mtb strains from smear-positive and smear-negative specimens [47]. Genotype MTBDRsl (Hain Lifescience) can detect mutations associated with fluoroquinolones and second-line drugs such as kanamycin, amikacin, and capreomycin [48,49].

#### 2.3.4. Micro Real-Time PCR

Truenat MTB, Truenat MTB Plus, and Truenat MTB-Rif Dx (Molbio Diagnostics) are micro real-time PCR-based assays for Mtb detection that produce results in 1 h [50,51]. Truenat MTB and Truenat MTB Plus detect Mtb bacilli in sputum after DNA extraction, and Truenat MTB-Rif Dx has an optional add-on chip for sequential RIF resistance detection [51]. In 2019, the WHO reported that the Truenat MTB series displayed comparable sensitivities and specificities with Xpert MTB/RIF and Xpert MTB/RIF Ultra for the detection of TB and RIF resistance [52].

Over the last decade, the development of molecular tests such as PCR has played a key role in the control of TB. These assays not only detect TB based on amplification of a targeted genetic region of the Mtb complex, but also detect the drug resistance of key drugs, such as RIF and INH. Importantly, molecular tests work more quickly than conventional Mtb culture and are also available at different levels of laboratories. Thus, they are becoming more and more important for TB diagnosis and are helping to improve the quality of TB care.

## 3. Immunological Diagnosis

### 3.1. Antibody Detection

Serologic tests rely on antibody recognition of Mtb antigens by the humoral immune response. Owing to the poor diagnostic sensitivity and specificity, the WHO does not recommend any commercial serologic assays for the diagnosis of TB, in case of misdiagnosis and resource waste [53].

### 3.2. Antigen Detection

The presence of circulating Mtb antigens can be detected from clinical specimens such as sputum, serum, and urine, based on the principle of sandwich enzyme-linked immunosorbent assay. Lipoarabinomannan (LAM) is a specific component of the cell envelope of Mtb and can be a potential biomarker for TB diagnosis [54]. FujiLAM is a urine lateral flow LAM test. The sensitivity and specificity of FujiLAM are 70% and 93%, respectively, in adult TB, while the sensitivity and specificity in children with TB are 51% and 87%. It performs better, with a higher diagnostic sensitivity, in patients with HIV infection or a CD4^+^ T cell count <200 cells/μL, both in adults and children [55].

### 3.3. Tuberculin Skin Testing (TST)

TST is a classical method based on detection of type IV hypersensitivity using purified protein derivative (PPD) of tuberculin. Mtb-infected patients can produce sensitized T lymphocytes with the ability to recognize Mtb antigens. When the sensitized T lymphocytes are stimulated by Mtb antigens again, a variety of soluble lymphokines are released to increase the vascular permeability, local redness, swelling, and induration [56,57]. The average diameter of induration is measured after 72 h of PPD injection as the results of TST. An average diameter of induration <5 mm or no response is considered as negative; ≥5 mm is considered as positive [58].

The following factors can influence TST results:

(I) Bacillus Calmette-Guerin (BCG) vaccination: Since BCG and PPD share antigenic components, the specificity of TST can be affected by BCG vaccination. The effect of BCG vaccination on TST in infancy is minimal, especially ≥10 years after vaccination [59]. BCG vaccination strategy (whether or not to multiply) also affects TST results, and the effect of BCG booster immunization on TST is more pronounced compared with the current BCG one-time vaccination [60].

(II) NTM infection: NTM is not a clinically important cause of false-positive TST, except in populations with a high prevalence of NTM infection and a very low prevalence of TB infection [59].

(III) The immune status of the host: in view of the fact that TST detection is based on Mtb-specific immune response, the immune status of the host will affect the accuracy of TST. Therefore, the sensitivity of TST for diagnosis of TB is reduced in patients with immunocompromised conditions [61]. A systematic review of the investigation of Mtb infection in immunocompromised populations showed that TST sensitivity decreased to 31% in hemodialysis patients [62]. Another study showed that immunosuppressed organ transplant recipients will likely develop anergy to the tuberculin antigen, which leads to false-negative TST results [63].

### 3.4. Interferon-Gamma (IFN-γ) Release Assays (IGRAs)

IGRAs are based on secretion of IFN-γ by lymphocytes exposed to Mtb-specific antigens (TBAg), such as early secreted antigenic target 6 (ESAT-6) and culture filtrate protein 10 (CFP-10) [64]. IGRA results are not affected by previous BCG vaccination and most infections caused by NTM, leading to a higher specificity than TST in detection of Mtb infection [64]. The two most common commercially available IGRAs are T-SPOT.TB (T-SPOT; Oxford Immunotec) and Quanti FERON-TB (QFT; Qiagen) [65,66]. 

#### 3.4.1. T-SPOT

T-SPOT assay, based on the enzyme-linked immunospot (ELISPOT) method, detects the number of IFN-γ-producing cells after Mtb-specific antigen stimulation. Currently, T-SPOT assay has been widely used for the diagnosis of Mtb infection [67]. T-SPOT has proven useful, not only in detecting Mtb infection in children and HIV patients [68,69], but also in the assessment of risk for TB development in chronic inflammatory diseases, prior to anti-TNF treatment and screening for latent tuberculosis infection (LTBI) in immigrant groups, health care workers, and college students [70,71,72]. T-SPOT has also been reported to be a useful adjunct test for diagnosing extrapulmonary TB [73]. In spite of the significance of the T-SPOT assay in diagnosing Mtb infection, the most critical limitation of this assay is its inability to distinguish active TB from LTBI [74,75,76]. Thus, this limitation led a WHO expert group to discourage the use of T-SPOT for the diagnosis of active pulmonary TB in low- and middle-income countries, because of an unsatisfactory specificity [77].

Our previous studies have expanded the application of the T-SPOT assay in the following three aspects:

(I) The ratio of TB-specific antigen (TBAg) to phytohaemagglutinin (PHA) (TBAg/PHA): the larger of the ESAT-6/PHA and CFP-10/PHA of T-SPOT assay is defined as the TBAg/PHA ratio [78]. We have found that calculation of the TBAg/PHA ratio of the T-SPOT assay can increase the specificity of this assay for diagnosis of active TB [78,79,80,81,82,83]. The theoretical basis is that TBAg/PHA ratio can eliminate the impact of individual immune variation on a T-SPOT assay. Specifically, active TB patients with immunocompromised conditions show decreased TBAg results, leading to increased difficulty in distinguishing active TB from LTBI, because low TBAg results are mostly attributed to LTBI. However, PHA results, the positive control of T-SPOT assay, are correspondingly decreased in these situations, because they can reflect the immune status of the host [79,80]. Thus, the TBAg/PHA ratio is still at a high level and better than directly using TBAg results in this condition. Conversely, LTBI individuals with high TBAg results may have much higher PHA results, leading to there still being a low TBAg/PHA ratio in this condition. Furthermore, very recently, we reported the potential value of TBAg/PHA ratio in the treatment monitoring of TB [84].

(II) Mean spot sizes (MSS): the MSS of ESAT-6 spot-forming cells in the T-SPOT assay is calculated with an automated ELISPOT reader (CTL Analyzers). Our study showed that the MSS of ESAT-6, but not CFP-10, of T-SPOT assay in active TB patients was significantly higher than that in LTBI individuals, supporting the evidence that the MSS of ESAT-6 can be used as an adjunct tool for diagnosis of active TB [85]. Expectedly, our findings demonstrated that a combination of the MSS of ESAT-6 and TBAg/PHA ratio of the T-SPOT assay showed potential in discriminating active TB from LTBI [85].

(III) Non-blood T-SPOT assay: non-blood samples, including pleural and peritoneal fluids, can also be used to perform T-SPOT assay. Our data showed that the performance of a peripheral blood T-SPOT in diagnosing TB pleurisy was limited, especially with a decreased sensitivity. However, using 1 × 10^5^ pleural fluid mononuclear cells for performing T-SPOT can improve the diagnostic accuracy of TB pleurisy, with a sensitivity and specificity of 89.76 and 96.70%, respectively [86]. In addition, our findings showed that, except for pleural and peritoneal fluids, other non-blood samples such as cerebrospinal fluid are not suitable for performing T-SPOT, due to it being difficult to harvest a sufficient number of lymphocytes for performing the experiment [87,88,89].

#### 3.4.2. QFT

A QFT assay is based on the enzyme linked immunosorbent assay to detect IFN-γ secreted into the supernatant of culture medium after Mtb-specific antigen stimulation. The advantages and limitations of QFT are similar to those of T-SPOT. The operational procedures of QFT are simpler than T-SPOT, as it does not require the isolation of peripheral blood mononuclear cells, but instead uses whole blood cells. However, the sensitivity of QFT for detecting Mtb infection is slightly lower than that of T-SPOT, and this is more pronounced in patients with immunocompromised conditions [90].

The fourth-generation QuantiFERON-TB Plus (QFT-Plus) assay, including another TB antigen tube that contains additional shorter peptides from ESAT-6 and CFP-10, is designed to detect both the CD4^+^ and CD8^+^ T cell responses [91]. It was developed with the hope of improving the detection of LTBI among immunocompromised hosts. However, studies comparing QFT-Plus to QFT currently do not support the superior performance of QFT-Plus in individuals with either active TB or LTBI [91,92,93,94].

In general, given that the immunological diagnosis of TB is based on detection of host antigen-specific responses, and not depending on Mtb load, immunological methods, especially the quantitative and high sensitivity methods like T-SPOT, have the potential for diagnosis of bacterial-negative TB. In addition, there is no doubt that the IGRAs are currently the best methods for screening LTBI, no matter whether in immunocompetent or immunocompromised individuals. Considering that Xpert MTB/RIF and TBAg/PHA ratio can make up for each other’s shortcomings based on detecting different aspects [95], we anticipate that a combination of these two methods may represent a good algorithm for prompt diagnosis of TB in highly endemic areas. We have summarized the characteristics of the typical methods for TB diagnosis in Table 1.

## 4. New Techniques

### 4.1. Next-Generation Sequencing (NGS)

Next-generation sequencing is considered a promising method for performing DST of TB and it produces results much faster than phenotypic culture-based testing [96,97]. Unlike probe-based assays that are limited to probe-specific targets, NGS can provide detailed and accurate sequence information for whole genomes by using whole-genome sequencing or multiple gene region sequencing. The WHO has published guidance on the role of NGS technologies for detecting mutations associated with drug resistance in Mtb complex [98]. Currently, some developed countries, such as the US and UK, have transitioned from phenotypic culture to whole-genome sequencing for DST for first-line drugs [99,100], and the US Centers for Disease Control sequences isolates from all culture-confirmed TB cases nationwide [101]. With the reduction of sequencing cost, NGS will be of importance for the surveillance of drug resistance of TB.

### 4.2. Mass Spectrometry

Routine matrix-assisted laser desorption ionization time-of-flight mass spectrometry (MALDI-TOF-MS) has proven to be useful for the identification of mycobacteria by direct analysis of deposits of a colony on MALDI-TOF-MS target [102,103]. Previous studies have shown that MALDI-TOF-MS is a fast and economical way to identify both Mtb complex and NTM species [104,105]. When an identification score of 1.3 is used, the positive predictive value of the identification of mycobacteria can reach up to 100% [106]. Given that MALDI-TOF-MS can provide results within a few hours and is faster than sequencing and hybridization-based techniques, it has potential as a rapid and reproducible method for the identification and typing of mycobacterium species.

Recently, a novel MALDI-TOF MS methodology, based on characterized species-specific lipid profiling of intact bacteria, was able to avoid a special treatment to bacteria for releasing molecules of interest. This method is fast (<10 min), highly sensitive (<1000 bacteria required), and specific for identification of Mtb complex strains [107]. Our preliminary study showed that serum the CFP-10 signal detected by nanotechnology and MALDI-TOF MS exhibited high diagnostic sensitivity and specificity for TB in infants and potential utility for monitoring anti-TB treatment [108]. In addition, some researchers used MALDI-TOF MS and liquid chromatography-tandem mass spectrometry (LC-MS/MS) to identify a TB-specific serum peptide profile and establish diagnostic models for rapid and accurate diagnosis of TB [109]. Nevertheless, these new methods still need further exploration and validation.

### 4.3. Artificial Intelligence (AI)

One of the main uses of AI in TB is using machine learning to automate the diagnosis of disease. The common strategy is to establish expert systems using a machine learning method based on the clinical, radiological, and laboratory data of TB patients. Interestingly, machine learning has been reported to aid clinicians in diagnosing pulmonary TB or predicting drug-resistant TB [110,111,112,113,114,115,116,117]. For instance, Lopes et al. presented three proposals for the application of pre-trained convolutional neural networks as image feature extractors to detect TB disease [110]. Jaeger et al. reported the possibility of discriminating automatically between drug-resistant and drug-sensitive TB in chest X-rays by means of image analysis and machine learning methods, using an artificial neural network in combination with a set of shape and texture features [112]. We also successfully developed a GBM model based on machine learning method, by using laboratory data independently, and this model may be of great benefit for serving as a tool in the identification of active TB [118]. Moreover, another area of AI driven interventions in a health context is morbidity and mortality risk assessment. Hussain et al. proposed a methodology using three machine learning algorithms, to facilitate TB programs by quantifying the risk of TB treatment failure, contributing to running TB programs more effectively [119]. Despite the potential role of AI in TB control, more studies should be carried out to validate the real performance of AI in clinical practice.

## 5. Conclusions

Mtb, one of the most successful pathogens in the world, has co-existed with humans for one thousand years and remains a major public health threat, causing over 2 million deaths annually [120]. An early diagnosis and effective treatment are the keys to controlling TB. Despite tens of thousands of research studies and clinical trials having been conducted, the early diagnosis of TB is still a challenge in clinical practice. Moreover, species such as urine and extrapulmonary tissue are sometimes difficult to collect, which also limits the performance of TB testing. Both under- and over-diagnosis of TB are inevitable, and empirical treatment is still a common strategy for diagnosis of clinical TB, especially in resource-limited countries. Therefore, the development of new strategies for diagnosis and prognosis of TB is a clinical problem that needs to be solved urgently, to end the global TB epidemic. Future research needs to focus on the following issues: (1) markers reflecting the status of Mtb infection, such as latent TB, preclinical TB, and active disease are lacking; (2) markers used to predict the development of active TB are lacking, because of the difficulty of conducting prospective studies; (3) markers used for monitoring the effect of anti-TB treatment are lacking; (4) methods used to classify the endotypes of TB are rare, while we believe that some types of TB can be recovered from without anti-TB treatment; (5) the new diagnostic methods that do not target the Mtb pathogen will play a more important role in the diagnosis of paucibacillary TB; (6) diagnostic methods which use non-respiratory samples are lacking; (7) the methods used for diagnosis of LTBI need to be further explored; (8) the immune characteristics of patients with LTBI need to be further explored; and (9) the differences in antigen-specific T cells between patients with active TB and LTBI need to be further explored.

## Figures and Tables

**Table 1 jcm-11-05826-t001:** Comparison of the typical methods for diagnosis of TB disease.

	Microscopy	Xpert MTB/RIF	Culture	T-SPOT.TB
Price	Low	High	Medium	High
Procedure complexity	Low	Low	High	High
Sensitivity	Low (for bacterial-positive TB)	High (for bacterial-positive TB)	High (for bacterial-positive TB)	Relatively high (for both bacterial-positive and bacterial-negative TB)
Specificity	High (in regions with a low incidence of NTM)	High	High	High (for diagnosis of Mtb infection), medium (for diagnosis of active TB in TB-endemic areas)
Advantages	Fast, simple, inexpensive	Fast, simple, low biosafety risk, detecting one drug resistance	Detecting all drug resistances	Detecting bacterial-negative TB, detecting latent TB infection
Shortcomings	Low sensitivity, cannot differentiate between live and dead bacilli	Expensive, cannot differentiate between live and dead bacilli	High complexity, long turnaround time, high biosafety risk	Expensive, high complexity

TB, tuberculosis; Mtb, *mycobacterium tuberculosis*; NTM, nontuberculous mycobacteria.

## Data Availability

All data is included in the manuscript.

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
