# Peer review of "Review and Updates on the Diagnosis of Tuberculosis"

_jcm, 2022, doi:10.3390/jcm11195826_

Round 1

Reviewer 1 Report

Diagnosis of tuberculosis and especially diagnosis of extrapulmonary tuberculosis, still faces challenges in clinical practice. There are several reasons for this. Methods based on detection of Mycobacterium tuberculosis (Mtb) are insufficiently sensitive, methods based on detection of Mtb-specific immune responses cannot always differentiate active disease from latent infection, and some of the serological markers of infection with Mtb are insufficiently specific to differentiate tuberculosis from other inflammatory diseases. New tools based on technologies such as flow cytometry, mass spectrometry, high-throughput sequencing, and artificial intelligence have the potential to solve this dilemma. This aim of this review is to provide an updated overview of current efforts to optimize classical diagnostic methods and new molecular and other methodologies for accurate diagnosis of patients with Mtb. (After the Abstract)

Tuberculosis remains a leading worldwide cause of death from infectious disease, second only to COVID-19 for the past few years. Ending tuberculosis (the current goal of WHO) is dependent on early, rapid and accurate identification of the infecting organism and determination of its drug susceptibility to enable initiation of optimal treatment and management. The authors have produced an appropriately extensive summary of currently available diagnostic tests for this disease as well as a brief but critical review of new diagnostic techniques in development that can soon be expected to be applied in the clinic. Attention to the following points would, however, be appropriate.

1. Abstract. This would benefit from a gentle revision (perhaps similar to the first paragraph above) to make it easier for the reader to understand exactly what the authors hope to accomplish in their review.

2. A number of small errors require correction.
a. line 9. Mtb should not be italicized.
b. line 143. This sentence should be revised to read, "It is based on hemi-nested real-time polymerase-chain-reaction (PCR)...".
c. line 144. The abbreviation TAT is undefined. It should be replaced by the phrase "turnaround time".
d. line 334. “Mass spectrum” should be replaced by "Mass spectrometry".  
e. lines 423-425. Line 425 should be deleted and line 424 revised to read "... Policy Statement. World Health Organization: Geneva, 2011." If this document is available online, the URL should be included. 

Author Response

We are grateful for the opportunity to submit a revised version of our manuscript (ID jcm-1915848) entitled “Review and Updates on the Diagnosis of Tuberculosis”. We greatly appreciate the insightful and constructive comments from reviewers. Those comments are valuable and very helpful for improving our paper. We have studied comments carefully and have made correction which we hope meet with approval. The responds to the reviewer’s comments are as following:

  1. This would benefit from a gentle revision (perhaps similar to the first paragraph above) to make it easier for the reader to understand exactly what the authors hope to accomplish in their review.

Response: Thank you for the comment. As suggested, we have revised the abstract to make it clear.

  1. Line 9. Mtb should not be italicized.

Response: Thank you for the comment. As suggested, we have revised the font of Mtb.

  1. Line 143. This sentence should be revised to read, "It is based on hemi-nested real-time polymerase-chain-reaction (PCR)...".

Response: Thank you for the comment. As suggested, we have revised this sentence.

  1. Line 144. The abbreviation TAT is undefined. It should be replaced by the phrase "turnaround time".

Response: Thank you for the comment. As suggested, we have replaced “TAT” with “turnaround time”.

  1. Line 334. “Mass spectrum” should be replaced by "Mass spectrometry".

Response: Thank you for the comment. As suggested, we have replaced “Mass spectrum” by “Mass spectrometry”.

  1. Line 423-425. Line 425 should be deleted and line 424 revised to read "... Policy Statement. World Health Organization: Geneva, 2011." If this document is available online, the URL should be included.

Response: Thank you for the comment. As suggested, we have revised this reference, and we have also revised references in line 402-405, 509-515, 523-525 and 663-666.

Reviewer 2 Report

Authors present an interesting review about previous and novel methods of TB diagnosis. Despite well-known TB course and available diagnosing techniques, fast diagnosis and efficacious TB treatment are hard to achieve. Although this review presents a lot of diagnostic procedures, I have minor concerns:

1) can you please add more data about difficulties in material collection for TB testing? i.ex. urine collection and No of samples;

2) please think about preparing a Table, in which all methods will be compared (pros and cons of each);

3) please be more specific, i.ex. in line 244 'transplant candidates' or rather 'transplant recipients' who have lower immune system responsiveness.

Author Response

We are grateful for the opportunity to submit a revised version of our manuscript (ID jcm-1915848) entitled “Review and Updates on the Diagnosis of Tuberculosis”. We greatly appreciate the insightful and constructive comments from reviewers. Those comments are valuable and very helpful for improving our paper. We have studied comments carefully and have made correction which we hope meet with approval. The responds to the reviewer’s comments are as following:

  1. Can you please add more data about difficulties in material collection for TB testing? i.ex. urine collection and No of samples;

Response: Thank you for the good comment. As suggested, we have added difficulties in material collection for TB testing as “Besides, the species such as urine and extrapulmonary tissue are difficult to collect sometimes, which also limits the performance of TB testing” (Last paragraph in the revised manuscript).

  1. Please think about preparing a Table, in which all methods will be compared (pros and cons of each);

Response: Thank you for the good comment. As suggested, we have added a table comparing typical methods for diagnosis of TB disease (Table 1).

  1. Please be more specific, i.ex. in line 244 'transplant candidates' or rather 'transplant recipients' who have lower immune system responsiveness.

Response: Thank you for the good comment. We are sorry that we blurred the concepts of transplant candidate and transplant recipient. To better express our point of view, we have revised this paragraph.

Reviewer 3 Report

This manuscript is quite a good topic and valuable for the entire world. 

Some minor revision is needed

Author Response

We are grateful for the opportunity to submit a revised version of our manuscript (ID jcm-1915848) entitled “Review and Updates on the Diagnosis of Tuberculosis”. We greatly appreciate the insightful and constructive comments from reviewers. Those comments are valuable and very helpful for improving our paper. We have studied comments carefully and have made correction which we hope meet with approval. The responds to the reviewer’s comments are as following:

  1. Line 96: Please write stand for NALC.

Response: Thank you for the good comment. As suggested, the abbreviation has been expanded.

  1. Line 206: Is it possible to use “Antibody detection”?

Response: Thank you for the good comment. As suggested, we have replaced “Serologic tests” with “Antibody detection”.

  1. Line 298: Is it possible to write down the "Quantiferon-TB (QFT)"?

Response: Thank you for the comment. We have described the abbreviation of Quanti FERON-TB (QFT) in the beginning of the 3.4.

  1. Line 323: Please replace with "Next-generation sequencing", because this is the beginning of the sentence.

Response: Thank you for the good comment. As suggested, we have replaced “NGS” with "Next-generation sequencing".